# Frequency Range of UHF PD Measurements in Power Transformers

Stefan Tenbohlen [1,*], Chandra Prakash Beura [1], Wojciech Sikorski [2], Ricardo Albarracín Sánchez [3], Bruno Albuquerque de Castro [4], Michael Beltle [1], Pascal Fehlmann [5], Martin Judd [6], Falk Werner [7] and Martin Siegel [8]

1 Institute of Power Transmission and High Voltage Technology (IEH), University of Stuttgart, 70569 Stuttgart, Germany
2 Institute of Electrical Power Engineering, Poznan University of Technology, Piotrowo 3A, 60-965 Poznan, Poland
3 Department of Electrical and Electronic Engineering, Automatic Control, and Applied Physics, School of Industrial Design and Engineering (ETSIDI), Universidad Politécnica de Madrid (UPM), Ronda de Valencia 3, 28012 Madrid, Spain
4 Department of Electrical Engineering, School of Engineering, São Paulo State University (UNESP), Bauru 17033-360, SP, Brazil
5 FKH Fachkommission für Hochspannungsfragen, 8050 Zürich, Switzerland
6 High Frequency Diagnostics and Engineering Ltd., Clydeway House, 813 South Street, Glasgow G14 0BX, UK
7 Doble Engineering Company, Marlborough, MA 01752, USA
8 BSS Hochspannungstechnik GmbH, 71229 Leonberg, Germany
* Correspondence: stefan.tenbohlen@ieh.uni-stuttgart.de

**Abstract:** Although partial discharge (PD) measurement is a well-accepted technology to assess the quality of the insulation system of power transformers, there are still uncertainties about which frequency range PDs radiate and which frequency range should be evaluated in a measurement. This paper discusses both a UHF PD frequency range obtained from studies investigating laboratory experiments and a frequency range from numerous practical use cases with online and on-site measurements. The literature review reveals a frequency spectrum of ultrahigh-frequency (UHF) PD measurements in the range of 200 MHz to 1 GHz for most publications. Newer publications extend this range from 3 to 6 GHz. The use cases present UHF PD measurements at transformers with power ratings up to 1000 MVA to determine frequency ranges which are considered effective for practical applications. The "common" frequency range, where measurements from all use cases provide signal power, is from approximately 400 MHz to 900 MHz, but it is noted that the individual frequency range, as well as the peak UHF signal power, strongly varies from case to case. We conclude from the discussed laboratory experiments and practical observations that UHF PD measurements in power transformers using either valve or window antennas, according to Cigré, are feasible methods to detect PD.

**Keywords:** power transformers; partial discharge; PD; UHF; monitoring; PD sensors; frequency range

## 1. Introduction

Ultrahigh-frequency (UHF) partial discharge (PD) measurement is a well-accepted technology to assess the quality of the insulation system of power transformers. For successful PD measurements, sensors and systems must be designed for the foreseeable measurable signals. Although 25 years ago the first publications on the UHF frequency spectra of PDs in oil showed significant emissions in the UHF range (300 MHz–3 GHz) [1], there are still uncertainties about which frequency range PDs radiate and which frequency range should be evaluated in a measurement. In gas-insulated switchgear (GIS), the UHF PD detection technique is already part of the acceptance test due to its good sensitivity and its possibilities for PD localization and classification [2]. Compared to GIS, in power transformers the discharges take place in an oil–paper insulation system instead of insulating

gases such as $SF_6$. Moreover, the geometry of the core, windings, and tank is not as regular as a GIS busbar, which acts as a coaxial waveguide for EM waves. These points sometimes lead to the incorrect statement during discussions that PD in oil cannot be detected by measurement technology in the UHF range.

In this paper, a selection of case studies is presented to demonstrate which frequency ranges for UHF PD measurements in transformers are effective in practical applications. A literature review of various laboratory measurements and the influencing factors on the frequency spectrum of PD pulses is presented. Then, examples of measurements on power transformers with ratings up to 1000 MVA are described and discussed.

## 2. Basics of UHF PD Measurement

The electrical method of PD measurement, according to IEC 60270, is significantly disturbed by external disturbances. Hence, this method cannot be used for continuous online monitoring. Even for the purpose of site acceptance tests (SATs), challenges posed by noise and interference can be prohibitive for conventional PD measurements due to the required sensitivity. PD monitoring using UHF sensors overcomes this issue in many on-site measurements since the PD signals are not galvanically coupled, and there are additional benefits, such as resilience to external noise and the possibility of three-dimensional PD source localization.

UHF PD monitoring is based on the principle that PD current pules are short-duration transients with fast rise times flowing through the winding of the transformer, causing the metallic surface of the winding nearby the PD source to emit electromagnetic (EM) radiation in the UHF range [3]. These EM waves propagate inside the transformer tank and can be detected by means of UHF sensors installed on the transformer tank wall.

Both the measurable electrical and the UHF PD levels are influenced by:

1. The type of PD source, including current pulse shape and its position in the tank.
2. The signal attenuation in the propagation path from the PD source to the detection system.
3. The sensitivity of the signal coupling device [4].
4. The sensitivity and frequency response of the measurement system.

### 2.1. Literature Review of Frequency Spectra in Laboratory Arrangements

The above-mentioned factors mean that obtaining reliable, unambiguous results of a frequency analysis of PD pulses is a complex task both for electrical and UHF PD measurements. This is confirmed by a review of research papers in which UHF PD pulses were analyzed (Table 1). Comparing the presented data, it can be seen how much the type of sensor (antenna) used, as well as the material and geometry of the test chamber, affect the parameters of the measured UHF spectrum.

**Table 1.** Influence of the type of antenna and test object on the frequency parameters of PDs.

| Ref. | PD Type | PD Pulse Frequency Range (MHz) | Peak Frequency (MHz) | Test Object | Antenna Type | Antenna Bandwidth (MHz) |
|------|---------|--------------------------------|----------------------|-------------|--------------|-------------------------|
| [1] | Floating electrode | 100–1200 | 700 | Metal test tank | N/A | 300–1200 |
| [5] | Needle in oil | 230–1380 | 211 | Metal test tank | N/A | 200–1500 |
| | Surface discharge | 380–1170 | 643 | | | |
| | Creeping discharge | 200–1300 | 200 | | | |
| | Turn-to-turn discharge | 1040–1300 | 1059 | | | |
| [6] | Needle in oil | 250–1100 | 490 | Metal test tank | Archimedean spiral antenna | 500–1500 |
| | Surface discharge | 450–930 | 465 | | | |
| | PD in oil wedge | 320–1050 | 575 | | | |
| | Void discharge | 250–680 | 425 | | | |
| | Floating metal particle | 260–850 | 290 | | | |

Table 1. *Cont.*

| Ref. | PD Type | PD Pulse Frequency Range (MHz) | Peak Frequency (MHz) | Test Object | Antenna Type | Antenna Bandwidth (MHz) |
|---|---|---|---|---|---|---|
| [7] | Needle in oil<br>Surface discharge<br>Floating metal particle | 5–960<br>150–1440<br>0–1000 | 35<br>378<br>370 | Dielectric test chamber | N/A | N/A |
| [8] | Needle in oil<br>Void discharge | 25–200<br>5–150 | 35<br>88 | Dielectric test chamber | Bi-conical/log-periodic antenna | 30–2000 |
| [9] | PD in oil wedge | 20–480 | 396/408 [1] | Metal test tank | N/A | N/A |
| [10] | Needle in oil<br>Surface discharge | 370–2540<br>275–1700 | 450<br>450/700 | Metal test tank [2] | Cone-shaped antenna | N/A |
| [11] | Needle in oil<br>Surface discharge<br>Floating metal particle<br>Void discharge | 260–775<br>260–760<br>260–765<br>260–400 | 349<br>518<br>525<br>302 | Metal test tank | Multi-band Peano fractal antenna | 345–600<br>660–735<br>920–1000 |
| [12] | Needle in oil<br>Surface discharge<br>Floating metal particle<br>Void discharge | 270–955<br>265–810<br>270–725<br>255–530 | 368<br>332<br>341<br>286 | Dielectric test chamber | Multi-resonant Hilbert fractal antenna | 300–1000 |
| [13] | Surface discharge<br>Void discharge | 300–950<br>310–940 | 370<br>425 | Dielectric test chamber | Meander antenna | 300–1000 |
| [14] | Floating metal particle<br>Void discharge | 400–735<br>270–620 | 695<br>335 | Metal test tank | Dual-arm logarithmic spiral antenna | 100–2000 |
| [15] | Surface discharge<br><br>Floating metal particle | 125–700<br><br>190–580 | 210<br><br>500 | Dielectric test chamber | Multi-band meander loop antenna | 480–520<br>800–850<br>1100–1150 |
| [16] | Surface discharge | 260–580 | 311 [3] | Metal test tank | Disc-shaped antenna | 150–1000 |
| [17] | Needle in oil | 20–1680 | 1140 | Dielectric test chamber | Wideband microstrip-fed planar elliptical monopole antenna | 1200–3000 |
| [18] | Surface discharge<br><br>Void discharge | 265–980<br><br>265–1000 | 900<br><br>605 | Dielectric test chamber | Multi-band stacked Hilbert antenna array | 339–375<br>395–440<br>450–1000 |
| [19] | PD in oil<br>Floating metal particle | 20–300<br>25–620 | 50 [4]<br>172 | Dielectric test chamber | Wideband helical antenna | N/A |

[1] depends on the position of the antenna; [2] without metal cover; [3] depends on the voltage value (311 MHz by 22 kV, 442 MHz by 26 kV, and 445 MHz by 30 kV); [4] first peak frequency (next peaks correspond to frequencies of 65 MHz, 112 MHz, and 127 MHz).

In addition to the measurements of frequency parameters using UHF antennas shown in Table 1, the spectrum of a PD at the sharp tip of a needle in oil is presented in a recent publication [20]. The PD pulse was directly measured by using a 70 GHz bandwidth oscilloscope. The PD current waveform was a very steep pulse with a rise time of a few tens of pico-second order. The measured rise times were 15.0 ps, 17.7 ps, and 27.3 ps. The authors determined the main frequency content of the PD pulses in the range of 3 to 6 GHz (5G Sub6 band), which suggests that the emitted frequencies in older publications may be underestimated due to the limited measurement bandwidth.

## 2.2. Types of PDs and Emission Spectrum

For the purposes of this article, the frequency spectra of six types of PDs, which are characteristic of the oil–paper insulation system of a power transformer, were investigated, i.e., PD in oil (PD1), surface discharge (PD2), creeping discharge (PD3), PD in oil gap (PD4), void discharge (PD5), and PD in gas bubbles in an insulating oil (PD6). To generate PD pulses, electrode systems were used, the schematic diagrams and dimensions of which are shown in Figure 1. The electrode systems were placed in a cylindrical test chamber with a

diameter of 150 mm and a height of 300 mm, made of organic glass, and filled with TRAFO EN mineral oil (Orlen, Plock, Poland).

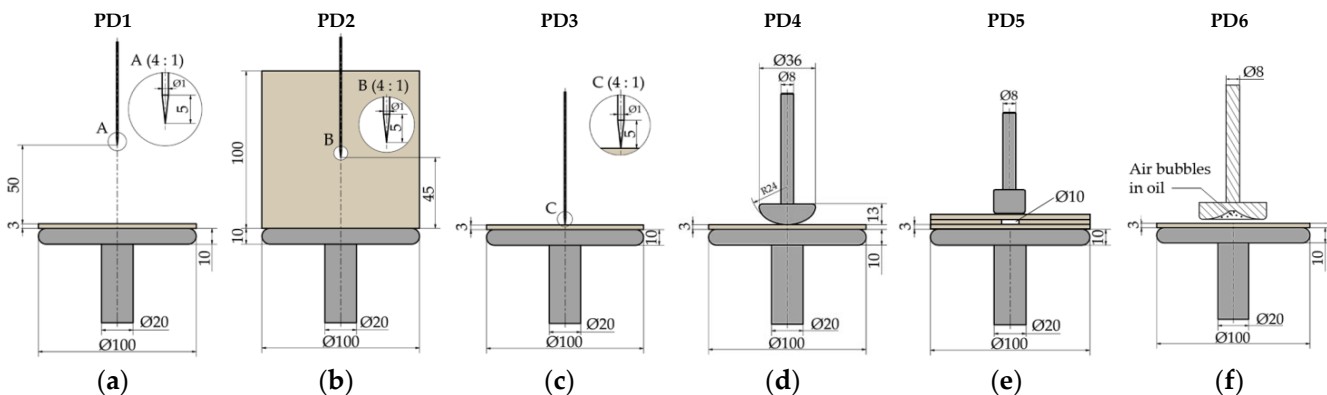

**Figure 1.** Schematic diagrams of electrode systems for generating PDs in oil–paper insulation: (**a**) PDs at needle tip. (**b**) Surface discharges on pressboard sample in oil with a negligibly small normal component of the electric field. (**c**) Creeping discharges (surface discharges on a pressboard sample in oil with significant normal component of electric field). (**d**) PDs in oil gap. (**e**) Void discharges. (**f**) PDs in air bubbles in mineral oil.

The measurement of PD in the UHF band was carried out in a shielded test chamber with dimensions of approximately 5.4 m × 2.9 m × 2.5 m, which correspond to the dimensions of the tank of a typical medium-power transformer. For the test chamber used, the theoretically determined main resonance frequencies for mode numbers m, n, p$\in$ {0, 1, 2} were in the range of 58 MHz for transverse electric mode $TE_{101}$ to 158 MHz for transverse magnetic mode $TM_{220}$ (see Section 2.3). To record the electromagnetic waves generated by PDs, a log-periodic VULP 9118A antenna was used, the frequency response of which covered a wide range from 150 MHz to 1500 MHz. In almost the entire range, the voltage standing wave ratio (VSWR) was less than 2. Considering both the resonant frequencies of the test chamber and the antenna frequency range, the recorded PD signals were filtered using a 150–1500 MHz bandpass filter. The antenna was placed at a distance of d = 2 m from the PD source, thanks to which the far-field condition was fulfilled, such that $d \geq 2L^2/\lambda$ where L is the largest dimension of the antenna aperture, and $\lambda$ is the wavelength [21].

UHF pulses from PD were acquired with a digital oscilloscope, MDO3104 (Tektronix Inc., Beaverton, OR, USA), with a 3 dB bandwidth of 1 GHz (6 dB bandwidth: 1250 MHz) and a sampling rate of 5 GS/s. According to the evaluation of the frequency response presented at the end of this section, only creeping discharges generated signals with a frequency slightly higher than 1 GHz, and the bandwidth is regarded as sufficient in this case. A high-frequency current transformer of the RFCT-4 type (Dimrus, Perm, Russia) was used to trigger the PD pulse acquisition process installed on the grounding conductor of the electrode system. In addition, the intensity of PD and the level of apparent charge q were monitored using a conventional PD meter, PD-Smart (Doble Engineering Company, Marlborough, MA, USA), according to the IEC 60270 standard. The basic parameters of the tested types of PDs are summarized in Table 2.

For each of the tested types of PD, at least 500 pulses were recorded, for which a frequency analysis was then carried out. Figure 2 shows normalized and averaged FFT spectra with values of standard deviation.

By analyzing the frequency spectra, it can be concluded that all tested PD types generated broadband radio signals. The creeping discharges (150–1150 MHz) have the widest band, and the PD in the oil gap has the narrowest band (150–400 MHz). The peak frequency did not exceed the value of 500 MHz, except for surface discharges, for which it was 503 MHz (Table 3).

**Table 2.** Parameters of the investigated types of PDs.

| Label | PD Type | PD Inception Voltage $U_i$ (kV) | Test Voltage Range $U_t$ (kV) | Max. Value of PD Apparent Charge $q_{max}$ (pC) | Mean Value of PD Apparent Charge $q_{mean}$ (pC) | Median Value of PD Apparent Charge $q_{med}$ (pC) |
|---|---|---|---|---|---|---|
| PD1 | PD at needle tip | 24.1 | 24.1–30.2 | 2 151 | 421 | 310 |
| PD2 | Surface discharges | 21.6 | 21.6–28.2 | 657 | 208 | 157 |
| PD3 | Creeping discharges | 11.9 | 11.9–24.0 | 12 971 | 1777 | 1305 |
| PD4 | PD in oil gap | 14.3 | 14.3–27.0 | 16 286 | 4273 | 3004 |
| PD5 | Void discharges | 11.6 | 11.6–24.6 | 278 | 151 | 134 |
| PD3 | PD in air bubbles in oil | 10.7 | 10.7–20.8 | 834 | 306 | 102 |

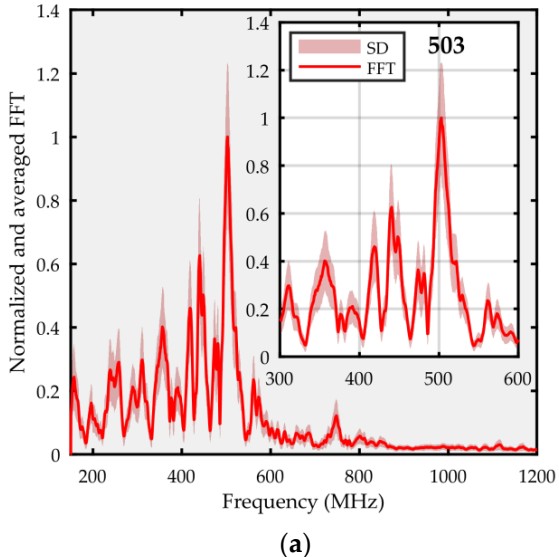 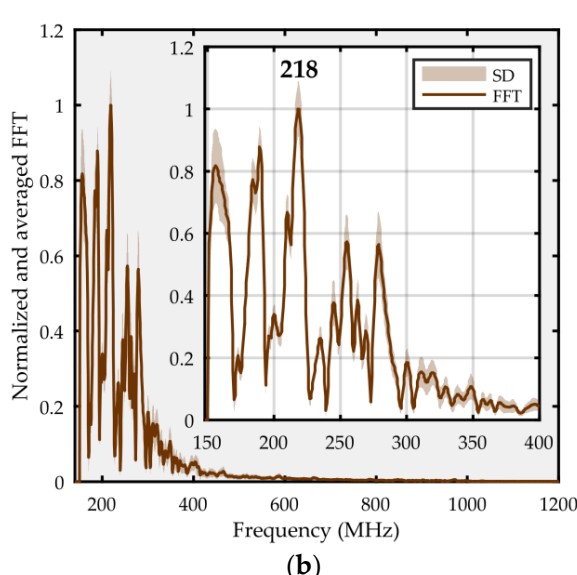

(**a**)                                                        (**b**)

**Figure 2.** Normalized and averaged FFT spectra with standard deviation (SD) values obtained for (**a**) surface discharges on pressboard sample in oil with a negligibly small normal component of the electric field; (**b**) PDs in air bubbles in mineral oil.

**Table 3.** Frequency parameters of the investigated types of PDs.

| Label | PD Type | PD Pulses Frequency Range (MHz) | Peak Frequency (MHz) | Test Object | Antenna Type | Antenna Bandwidth (MHz) |
|---|---|---|---|---|---|---|
| PD1 | PD at needle tip | 150–820 | 170 | | | |
| PD2 | Surface discharges | 150–865 | 503 | | Wideband log-periodic VULP 9118 A antenna | |
| PD3 | Creeping discharges | 150–1150 | 311 | Dielectric test chamber [1] | | 150–1500 |
| PD4 | PD in oil gap | 150–400 | 237 | | | |
| PD5 | Void discharges | 150–520 | 262 | | | |
| PD6 | PD in air bubbles in oil | 150–470 | 218 | | | |

[1] Oil-filled acrylic chamber with electrode system for PD generation was placed inside a shielded test chamber.

### 2.3. Signal Attenuation within Transformer Tank

Besides the natural radiation from the PD, the measurable UHF spectrum of a PD in a transformer depends strongly on the different transfer functions included in the propagation path from the source to the measuring system. In addition to the sensor and cable characteristics, the surrounding materials and objects, with their properties of transmission, scattering, and reflection, have a strong impact on the appearance of the recorded UHF PD signal [10].

The empty transformer tank without an active part can be considered a rectangular resonance cavity [10]. The radio waves propagating in this cavity are repeatedly reflected, creating transverse magnetic (TM) and transverse electric (TE) standing waves between the conductive walls, which in turn leads to the phenomenon of resonance. The resonant frequencies for the TM and TE modes in a rectangular cavity are defined as

$$(f_r)_{nup} = \frac{c_0}{2\pi\sqrt{\mu_r\varepsilon_r}}\sqrt{\left(\frac{m\pi}{a}\right)^2 + \left(\frac{n\pi}{b}\right)^2 + \left(\frac{p\pi}{c}\right)^2} \tag{1}$$

where $f_r$ is the resonant frequency of a rectangular cavity with dimensions $a \times b \times c$, $c_0$ is the speed of light in vacuum, and $\mu_r$ and $\varepsilon_r$ are the relative permeability and relative permittivity of the cavity-filling medium, respectively. In the case of transverse electric (TE$_{mnp}$) modes, $m \wedge n \in \{0, 1, 2, \ldots\}$, $p \in \{1, 2, 3, \ldots\}$, and at least two modes have to be non-zero. For the transverse magnetic (TM$_{mnp}$) modes, $m \wedge n \in \{1, 2, 3, \ldots\}$ and $p \in \{0, 1, 2, \ldots\}$. To real transformers with many internal metal structures, this kind of analysis is less relevant due to the attenuation of the coupling path.

In an experimental analysis, monopole antennas were installed on the walls of two 300 MVA transmission transformers (Tr A and Tr B) and used to send and receive artificial UHF PD signals [22]. The signal attenuation with respect to the distance between source and receiver was analyzed for both transformers. For analysis of the signal behavior, three types of propagation were defined: namely, direct, indirect, and lateral. When the source and receiver were on the same tank wall, the propagation was considered to be "direct", as there should be minimal obstacles to signal propagation. When the source and receiver were on opposite tank walls, the signal propagation was considered to be "indirect", as the EM waves would have to propagate around the active part. Lastly, the third category, "lateral", was used when the source or receiver was on the side wall of the transformer.

The measurement data obtained from both transformers were separated on the basis of the three aforementioned types of signal propagation paths, as shown in Figure 3. Similar conclusions can be drawn for both transformers, namely that the signal attenuation is higher at comparable line-of-sight (LoS) distances when the propagation is indirect, thus demonstrating the effect of the active part on signal attenuation. Additionally, the signal attenuation is similar in the case of direct propagation and lateral propagation, indicating that lateral propagation is also direct.

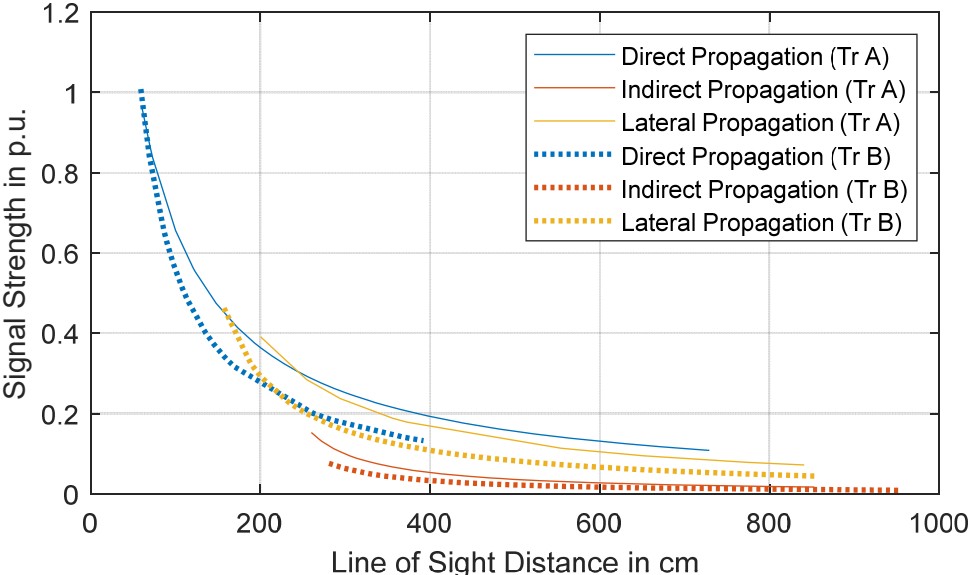

**Figure 3.** Comparison of signal attenuation based on the type of propagation in transformers A and B [22].

### 2.4. Sensor Sensitivity

Based on the installation mechanism, UHF sensors can be classified into two types: drain valve sensors and window-type sensors. The former is directly inserted into the oil volume of the transformer tank using a DN 50 or DN 80 drain valve. The latter is installed on a dielectric window, which is available on the tank wall, i.e., the sensor is not immersed in oil [23]. Other differences include the insertion depth of the sensor, which can be varied in the former but is fixed in the case of the latter.

Different types of antenna designs can be used for the aforementioned types of sensors. The most used designs are monopole, Archimedean spiral, and Hilbert fractal antennas, where monopole geometries are mostly used for practical applications and commercial UHF antennas. The monopole is a resonant antenna and functions as an open resonator for UHF waves, oscillating with standing waves of voltage and current along its length. Therefore, the length of the antenna is determined by the wavelength of the UHF waves it is used with.

Antenna-type sensors are described by different characteristics, e.g., by the antenna gain or the antenna aperture. For sensors which are not defined by a physical area, such as monopoles or dipoles, the effective height $H_e$ [24] or the antenna factor $AF$ can be used. $AF$ is defined as a function of signal frequency $f$ as follows [23]:

$$AF(f) = \frac{E(f)}{U(f)} \tag{2}$$

where $U(f)$ is the voltage at the sensor terminals (with 50 Ω load), and $E(f)$ is the electric field strength incident on the sensor (with the electric field vector aligned in the direction of the sensor axis). The antenna factor of a drain valve UHF sensor measured in an oil-filled GTEM cell up to 1.2 GHz is shown in Figure 4 [25]. A low antenna factor provides higher sensitivity, meaning lower is better. At higher frequencies (not shown in Figure 4), the antenna factor is higher (less sensitive).

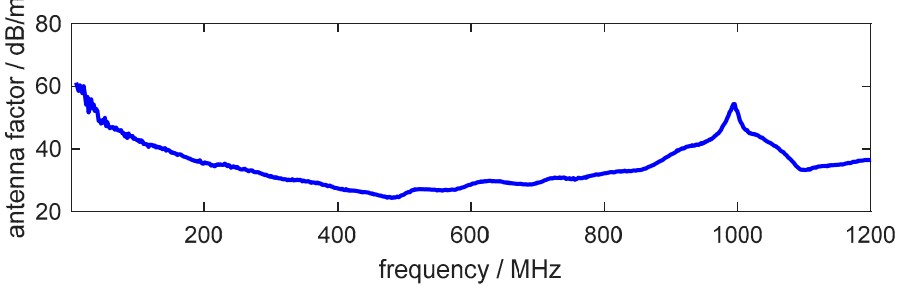

**Figure 4.** Antenna factor of the used sensor according to Cigré TB 861 [23].

### 2.5. Sensitivity of Measuring Device

Of course, the system bandwidth of the measuring device, including accessories such as cables, filters, attenuators, preamplifiers, etc., also influences the measurement result. For the analysis of the frequency spectra emitted by PD, the system bandwidth should not be lower than 1 GHz. As Cigré TB 861 [23] discusses and the use cases in this contribution show, there is a high probability of UHF signals in the frequency range of a few 100 MHz to approximately 1 GHz. However, a higher bandwidth can be beneficial in individual cases where a UHF signal power above 1 GHz is present.

For a calibrated UHF PD measurement, this influence is addressed by introducing the calibration factor $K_M$. Additionally, a measurement instrument needs to satisfy two pre-conditions:

(1)     A linear behavior for the complete dynamic range.
(2)     Frequency independence over a specified frequency range [23].

## 3. Use Cases

### 3.1. Laboratory Experiments

PD measurements using a 5 GS/s Tektronix DPO 7254 oscilloscope with monopole antennas performed inside and outside a model of a transformer tank show that the spectrum of the received signals is influenced by several factors. The first is the cavity, which, depending on its dimensions, supports certain resonant modes that maximize certain signal frequencies. The second is the frequency response of the antenna. The third is a dependence on the type of PD source. The model tank acts as a low-pass filter with a cut-off frequency of about 1.2 GHz so that an outer antenna is not able to receive power from the PD above this frequency. Furthermore, it was found that there may be a higher power content outside than inside, below 300 MHz, and this is attributed to the ground cable of the enclosure, which behaves as an antenna itself that accentuates the PD radiation below 300 MHz in the very high frequency (VHF) range. When the antenna is located outside the tank, a similar power below 300 MHz is measured both inside and outside, which could be due to the insufficient EM enclosure of the tank model. In this case, as with other test objects inside the enclosure that produce surface PD in a twisted pair configuration, resonance modes of the tank are excited, and the cumulative power has the same order of magnitude both inside and outside in the 500–1200 MHz range. These results suggest that, when considering only the power spectrum of the PD, it is not easy to identify the type of PD source. Therefore, it is preferred to have a voltage phase reference to represent the PRPD pattern to identify the type of source in each case.

Figure 5a shows an internal PD pulse for Nomex submerged in mineral oil measured by both antennas, one inside and another outside the tank. Their spectra compared to the background noise are, respectively, depicted in Figure 5b,c inside and outside the tank. As might be expected, the signal from the inner antenna has more content in the UHF band than the signal measured outside the enclosure, which is mitigated due to the loss of its energy when the pulse reverberates inside the tank [26].

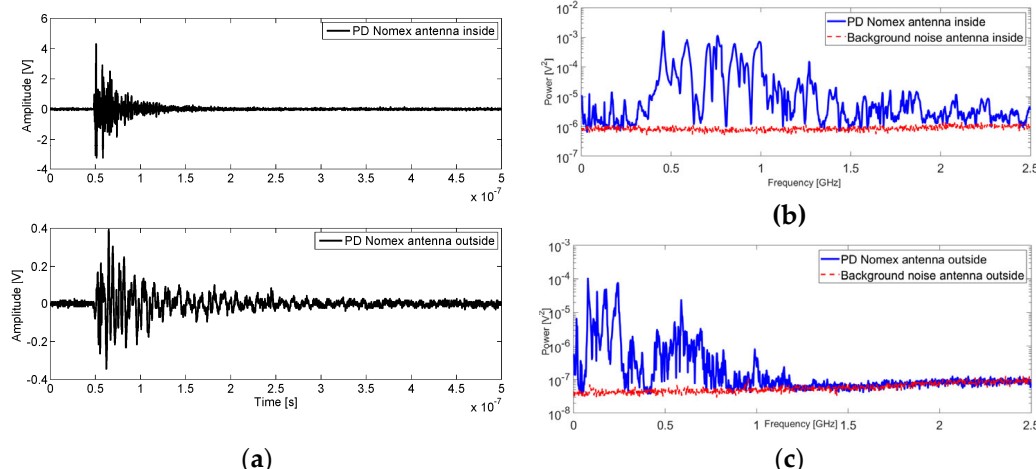

**Figure 5.** (**a**) Internal PD pulse for Nomex measured by both antennas; (**b**) frequency spectrum of the PD pulse compared to background noise inside the tank and (**c**) outside the tank.

### 3.2. Distribution Transformer (30 kVA, 13.8 kV/220 V)

Interturn short circuits (ITSCs) or PDs on bushings can impair the operation of the power transformer and lead the asset to total failure due to dielectric degradation. ITSCs can start due to high inrush currents and overload operational conditions, which can stress dielectric materials, creating spots without insulation. Mechanical issues, such as winding conductor bending, axial instability, careless transportation of transformers, vibrations, and winding movement due to loosened clamping structures, can also generate spots of insulation losses and start the full-discharge activity [27].

Bushings are constantly subjected to surface contamination by dust, moisture, hot spots, and porcelain damage due to electrical flashovers, lightning, cracks in outer coatings, etc. All mentioned issues are characterized as material degradation since the original properties of the component are weakened [27].

In this context, the frequency content of the UHF signals produced by an ITSC and bushing PD failures was assessed in an oil-filled power transformer (30 kVA, 13.8 kV/220 V). A bushing was contaminated with powdered graphite since, in real scenarios, external insulation systems are frequently contaminated with dust, which can promote surface PD activity. To emulate the ITSC, an electrode with a gap of 5 mm was immersed in the transformer's oil. The position of the electrode was the medium point of the front of the core. One hundred signals of surface PDs at the bushing and one hundred signals of full discharge were acquired. The goal was to perform a frequency analysis of the UHF signals emitted by the ITSC and PDs and perform failure classification since flaws require different maintenance actions.

Description of measuring equipment and measured signals:

Aiming to assess UHF signals emitted by the ITSC and bushing PDs, a Vivaldi antenna was installed two meters from the power transformer. The 8 bit high-speed digitizer M4i.2233-x8-SPECTRUM Instrumentation recorded the UHF time-domain signals with a sampling rate set at 4 GS/s. Figures 6 and 7 show the UHF signals of each failure type in the time and frequency domain, respectively.

It can be noted that the bushing discharge spectrum was concentrated after 400 MHz and presented a peak between 400 MHz and 600 MHz. The frequency band of ITSC failure was prevalent before 800 MHz, being a more distributed spectrum than the other flaw.

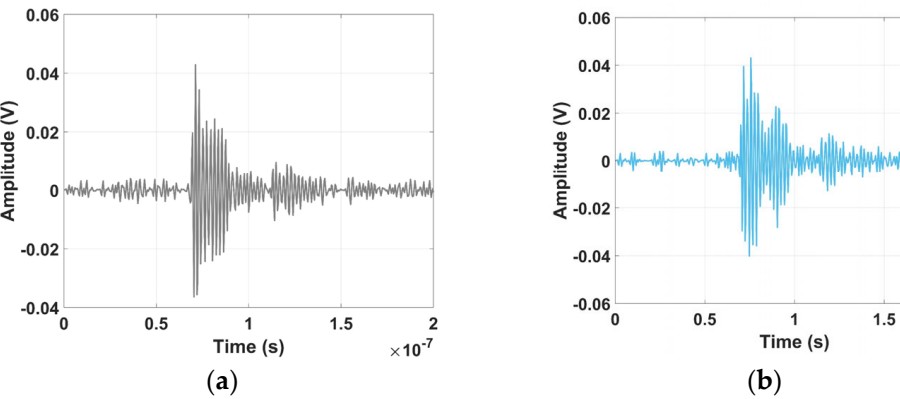

(a)                                              (b)

**Figure 6.** (**a**) Time-domain signal of a PD in bushing; (**b**) interturn short circuit.

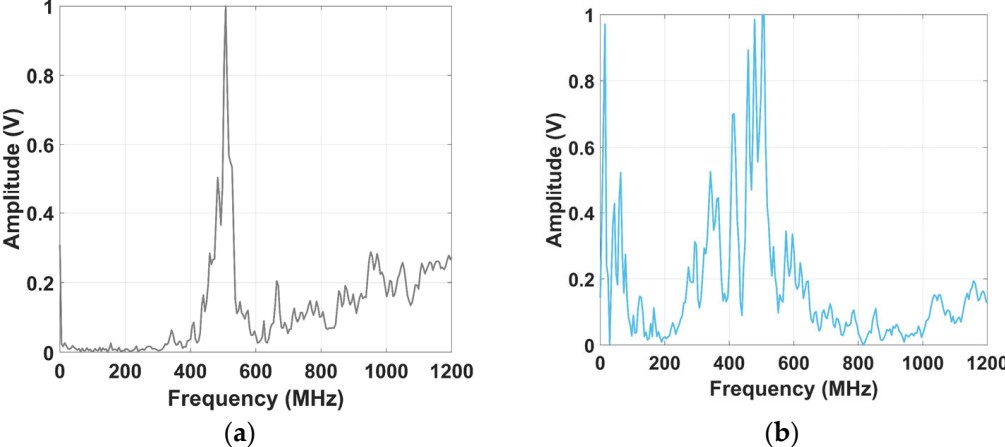

(a)                                              (b)

**Figure 7.** (**a**) Spectrum of a PD in bushing; (**b**) interturn short circuit.

### 3.3. Gassing of Traction Transformer 18 MVA

An 18 MVA 132/25 kV single-phase transformer forming part of a rail network supply was fitted with a UHF PD monitoring system during an outage, prompted by increasing rates of hydrogen and acetylene generation in its dissolved gas analysis. Three inspection hatches on top of the tank were modified so that each accommodated a UHF window sensor. When the transformer was re-energized, large UHF signals were recorded using a digital oscilloscope (10 GS/s, 3 GHz bandwidth). No preamplification was necessary because the signals were typically 50–100 mV peak to peak [28]. A typical set of UHF signals acquired during testing is shown in Figure 8. These were acquired for PD location purposes, but the present context focuses on the frequency content. Performing an FFT on these three signals provided the relative spectral density plots of Figure 9, which show that most of the PD signal energy is at frequencies above 500 MHz.

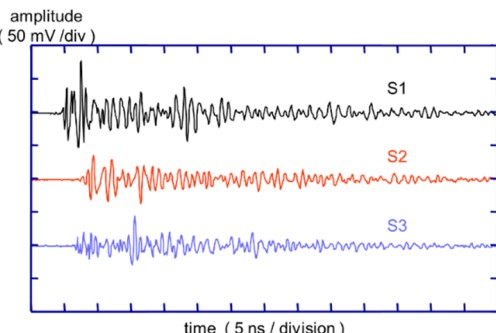

**Figure 8.** Time-domain signals captured simultaneously at 3 different UHF sensors S1–S3 for a single PD pulse in the 18 MVA transformer.

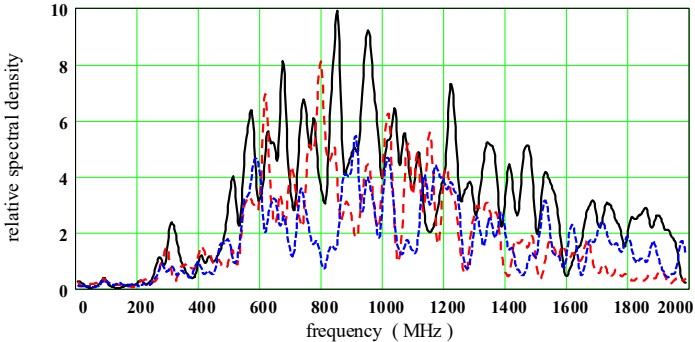

**Figure 9.** Comparing frequency spectra of the 3 PD signals shown in Figure 8.

### 3.4. Gassing of a 33 MVA, 132 kV Substation Transformer

The transformer in this example exhibited a significant amount of hydrogen in the oil, triggering a PD assessment. The spectral analysis, acquired with a Doble DFA-300 using a DN-80 drain valve antenna, is shown in Figure 10. No amplification was used.

The transformer showed significant PD activity, which was observed across the entire spectral range of the detector between 50 MHz and 1 GHz. Phase-resolved PD patterns indicated the presence of void discharges within the insulation system. An acoustic location attempt did not yield useful data.

The assessment of this transformer also included PD measurements on the cable terminations, which utilized high-frequency current transformers (HFCT) on the ground connections, in a frequency range limited by the sensor, of up to 300 MHz. In these measurement locations, contact issues and delamination-type PD were observed. The PD emissions detected in the main tank via the DN-80 antenna were not observed in these locations. Similarly, the PD observed on the cable terminations using the HFCTs was not observed in the main tank measurement.

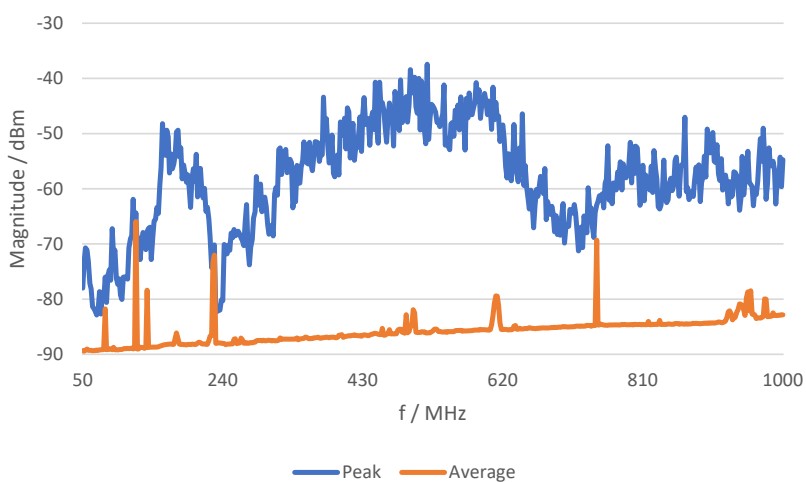

**Figure 10.** Spectrum Acquired on DN-80 Antenna.

### 3.5. Factory Acceptance Test (FAT) of 40 MVA, 220 kV Substation Transformer

A newly built three-phase transformer 220 kV/63 kV showed constant PD during a factory acceptance test using conventional measurement methods. It was a relatively small transformer with tank dimensions of approximately 5 m × 2 m × 3 m. To locate the PD source for direct factory repair, a combination of UHF and acoustic measurements was performed. One access for a DN 50/80 drain valve UHF sensor was available (a performance check was not possible).

After installation of the sensor, the applied voltage was slowly increased on the transformer during an induced voltage test, and UHF PD signals were recorded in the time domain using a 4 GHz, 40 GS/s, 8 Bit oscilloscope without preamplification. UHF was used as the trigger for the acoustic sensors. Localization was successful, and the PD source was found near the lead exit and could be repaired in the factory. Figure 11 shows an exemplary UHF time-domain signal measured at the transformer. Also shown is the frequency domain from the very same signal, which provides frequency contents from approximately 400 MHz up to 1.6 GHz. As the diagrams show, signal strength is quite high, with a satisfactory signal-to-noise ratio. The results are supported by the location of the source at the lead exit (with direct signal propagation into oil space and not inside a winding with accordingly higher signal attenuation) and the relatively small dimensions of the tank.

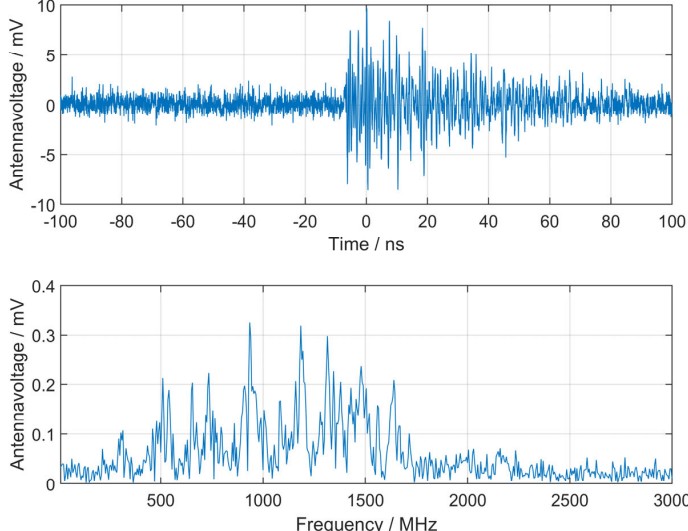

**Figure 11.** Time-domain signal and frequency spectrum of the same signal obtained from the installed sensor.

### 3.6. PD Location at 40 MVA, 140 kV Substation Transformer

A 40 MVA, 140 kV/15 kV transformer showed PD during a conventional electrical PD test during the factory acceptance test. To localize the PD, a combination of drain valve UHF sensors and tank-mounted acoustic sensors was used. Two drain valves were accessible, and both were used to install UHF sensors.

Broadband measurements in the time domain were performed for both installed UHF sensors using a digital sampling oscilloscope (DSO) with a 4 GHz analog bandwidth, 40 GS/s sampling rate, and 8 bit vertical resolution without preamplification. Figure 12 shows both the time-domain signals of single-shot measurements and the frequency-domain signal of the same.

The UHF signals had a relatively small bandwidth, mainly containing frequencies at approximately 600–900 MHz for sensor 1 and below 500 MHz for sensor 2. In the time-domain measurement on the left side in Figure 12, UHF sensor 1 was closer to the PD source and therefore has a steeper flank and higher level than UHF sensor 2. This corresponds in the frequency domain to higher frequencies for UHF sensor 1 and lower frequencies for UHF sensor 2. By this characteristic signal attenuation in the propagation path, a rough estimation of the PD location can be made, even without having enough sensors installed for time-of-flight algorithms.

In this case, UHF-triggered acoustic localization using the time-of-flight algorithm as described in [29] accurately located the PD source, and the transformer was repaired with minimal effort. The following subsequent FAT with conventional PD measurement was passed.

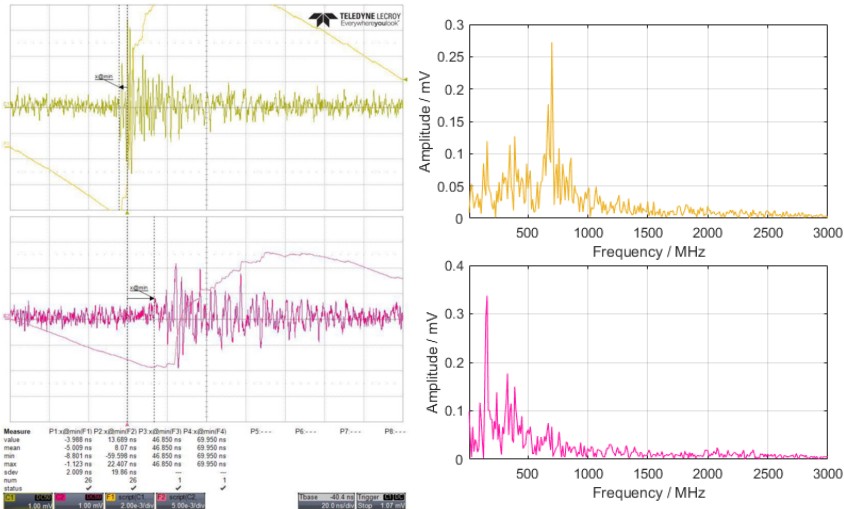

**Figure 12.** Screenshot of the time-domain signals (**left**) measured with the DSO, and corresponding frequency-domain signal (**right**) created with FFT.

### 3.7. Online Monitoring of Repaired 40 MVA, 123 kV Substation Transformer

After internal flashover, the active part of a 15-year-old 123 kV substation transformer was cleaned from deposits and carbon particles. After the repair, an induced voltage test with partial discharge measurement (IVPD) according to IEC 60076-3 was performed with a test level of 80% of the original test level. Since significant PD was detected during the factory acceptance, the transformer was equipped with an online PD monitoring system to assess the insulation condition on a permanent basis.

For the UHF PD monitoring, two UHF antennas of type Alstom MS3000 were installed on two DN80 oil gate valves. Sensor 1 was mounted on the oil drain valve at the bottom of the tank, and sensor 2 was mounted on the oil valve on top of the tank. The antennas were installed with an insertion depth of 50 mm for sufficient sensitivity.

During the online monitoring, PD activity was detected by both UHF antennas, and signals received by the antennas were measured using a LeCroy WaveRunner 640Zi Digital Storage Oscilloscope (DSO) using coaxial cables. The DSO has the following specifications: a 4 GHz analogue bandwidth, 10 GS/s sample rate (4 channels), 8 Bit vertical resolution, and 50 Ω input impedance.

The signal measured by sensor 1 is shown in Figure 13. In the frequency domain, it can be observed that there are components of the signal between 170 MHz and 1.2 GHz. Even though there are signal components below the UHF range (300 MHz), most of the signal lies in the range of 300 MHz–900 MHz.

In Figure 14, the signal received by sensor 2 during the same PD event is shown. In the time domain, compared to the signal shown in Figure 13, it can be observed that the signal is noisier and has a lower signal strength and longer time of arrival, which implies that the PD source is farther away from sensor 2. In the frequency domain, the signal has components between 70 MHz and 900 MHz. Again, even though there are components of the signal below the UHF range, the highest signal strength lies in the range of 300–600 MHz. The frequency range above 600 MHz is considerably more attenuated than with sensor 1, which may be due to the greater distance from the PD source.

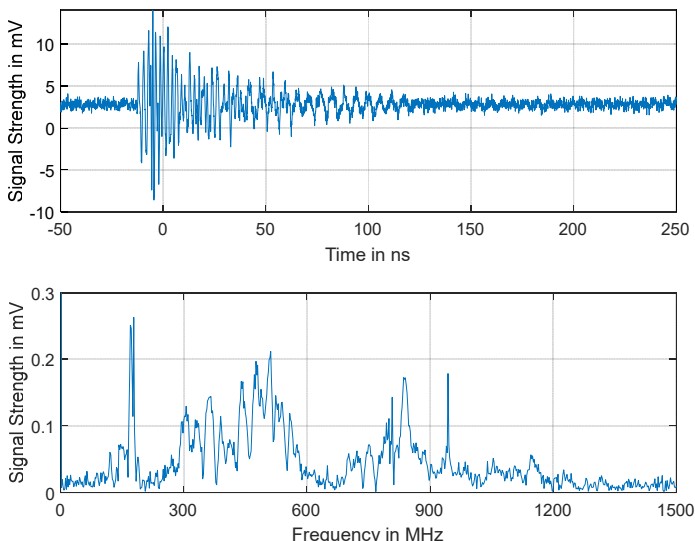

**Figure 13.** Time-domain signal and frequency spectrum of the same signal obtained from sensor 1.

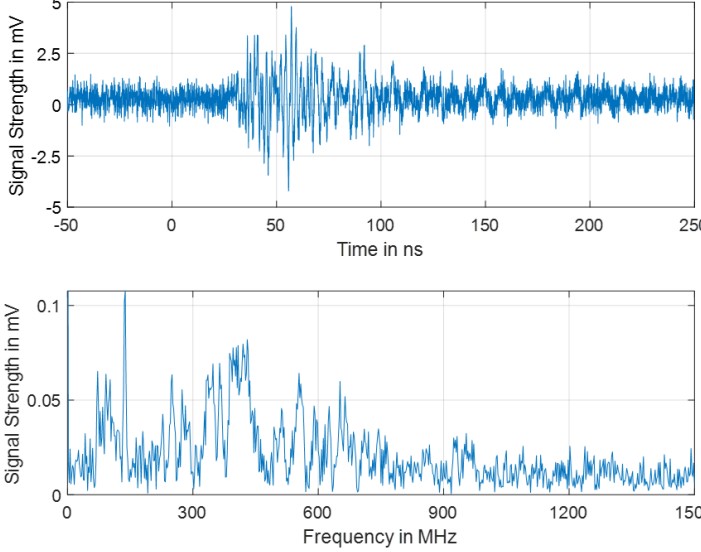

**Figure 14.** Time-domain signal and frequency spectrum of the same signal obtained from sensor 2.

### 3.8. On-Site PD Localization of 333 MVA, 420 kV Single-Phase Substation Transformer

Because of increasing gas-in-oil values, a 400/220/30 kV, 333 MVA substation single-phase autotransformer was tested on-site and online for PD. The high noise level at the site strongly disturbed the conventional PD measurements, made according to IEC 60270, below 1 MHz frequency. Consequently, UHF PD measurements for PD detection in combination with acoustic measurements for later PD localization were performed in order to get reliable results [29].

Frequency analysis of the measured signals of the installed UHF probes proved the shielding characteristic of the tank (see Figure 15). The signal features frequency contents of up to 1 GHz, as emitted by a broadband emitter of UHF waves, such as internal PD in oil. External disturbing sources would have been narrow banded, e.g., at around 500 MHz for digital video broadcasting or around 900 MHz or 1800 MHz for GSM since there are often modulated carriers. In Figure 15, the unamplified measured signals of the UHF probes are shown with their frequency analyses (FFT).

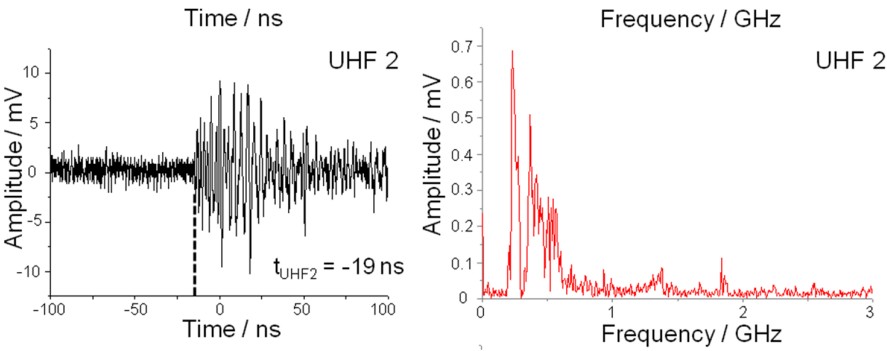

**Figure 15.** Measured UHF signal of probe "UHF 2" to prove broadband emission [29].

### 3.9. Examples from SATs

After completion of the on-site installation work of transformers, in some cases the transformer operator requires an IVPD measurement as part of the SAT. With these tests, dielectric faults in the insulation should be excluded. In addition to the "conventional" established PD measurement via the measuring tap of the bushings, UHF PD sensors are sometimes used.

For UHF PD measurements, a spectrum analyzer is used with a frequency range from 100 MHz and 2 GHz to analyze the signal (experience has shown that a frequency range between 300 MHz and 750 MHz is sufficient). If in some areas of the frequency spectrum a signal above the basic noise level is detected, a narrow-band filter is applied with a center frequency at the location where the signal is most clearly visible. Phase-resolved partial discharge analysis (PRPDA) diagrams are used to distinguish between phase-correlated PD activity in the tested transformer and external disturbance. Figure 16 shows the measurement principle.

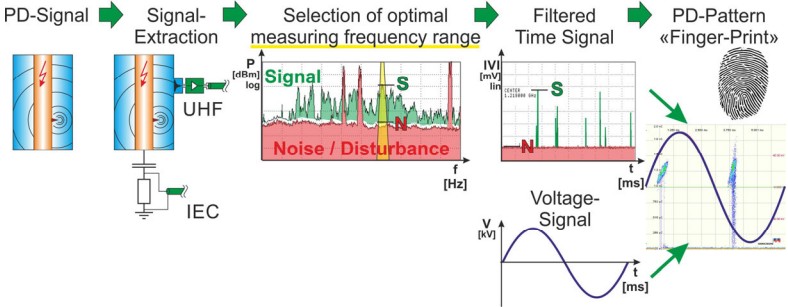

**Figure 16.** UHF measurement principle with a spectrum analyzer [30].

In both examples of UHF PD measurements below, the following measuring devices were used:

- UHF sensor in example 1: window-type UHF-PS1, BSS.
- UHF sensor in example 2: window-type UHT1, Omicron.
- Preamplifier: SMT-55-0018, Swissmains.
- Multiplexer: RF Multiplexer with integrated bias tee power supplies, Swissmains.
- Spectrum analyzer: Agilent 8594E, 9 kHz–2.9 GHz.
- PRPDA with Omicron MPD600 via the video output of the spectrum analyzer.

3.9.1. SAT on a One-Phase Phase-Shifting Transformer, 267 MVA, $U_m$ 420/245/36 kV

Figure 17 shows the measured frequency spectrum. The UHF sensor was installed on the front side facing the bottom corner of the tank (on this side, the HV bushing is located). The PRPD pattern was recorded with a "conventional" PD measurement, and the pattern detected by the UHF PD measurement suggests that both signals were created by the same PD source.

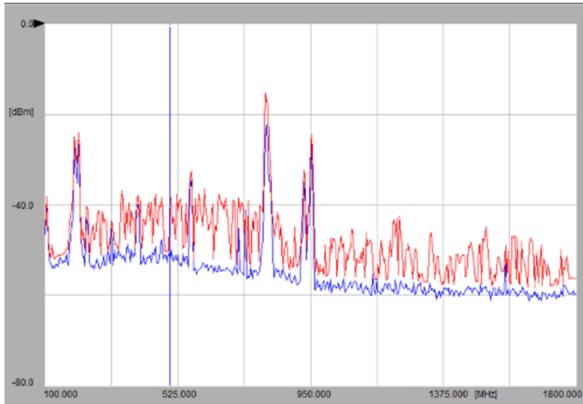

**Figure 17.** Frequency spectrum measured at the UHF sensor (blue curve: ground noise, red curve: PD signal, blue vertical marker at 500 MHz).

3.9.2. SAT on a Three-Phase Grid Transformer, 100 MVA, $U_m$ 300/72.5 kV

During the SAT of a three-phase grid transformer, a PD measurement was performed simultaneously at all HV and LV bushings and using four UHF sensors, which were placed in the corners of the transformer housing (two at the bottom, two at the top).

With the conventional PD measurement on the bushing taps, no PD signal was measurable either at the HV or at LV side. However, a low PD signal was detected on three of the four UHF sensors. Figure 18 shows the measured frequency spectrum at one of the UHF sensors.

These two examples demonstrate that UHF PD measurement is a valid method for the detection of PD in transformers. Depending on the location of the PD source, the UHF method may also be more sensitive than the established PD measurement at the bushing taps. At the same time, UHF PD measurement is less sensitive to external disturbances (which is an advantage for online PD measurements). This fact should be considered when interpreting measurements on transformers in operations which were installed years ago. In many cases, it cannot be excluded that a PD source detected with UHF sensors on aged transformers (e.g., in an investigation following a suspicious DGA) has been present since commissioning but was not detected with the "conventional" PD measurement during FAT back then.

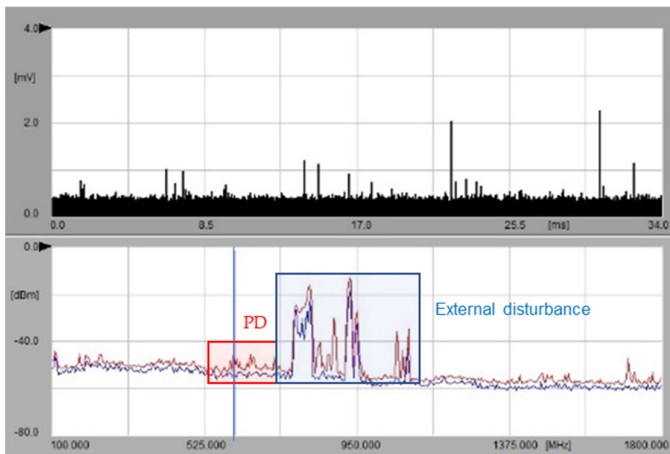

**Figure 18.** Frequency spectrum measured at the UHF sensor (black curve: PD signal in the time domain (zero span), blue curve: ground noise, red curve: PD signal, blue vertical marker at approximately 610 MHz).

*3.10. Gassing of a Single-Phase Autotransformer 243 MVA*

PD assessment was performed in early 2018 on a 243 MVA autotransformer in the wake of increased DGA levels indicating localized overheating. The purpose of the assessment was to determine the approximate geometric location of a potential defect within the transformer. The unit was built in 2003 and showed a sudden change in gas generation patterns evident in the DGA starting in May 2016.

The measurement instrument for UHF and acoustic emission tests was a Doble DFA-300, which provides basic UHF and acoustic emission testing capabilities. Additionally, a four-channel acoustic emission analyzer was utilized for geometric location purposes. For the UHF PD tests, a DN-58/80 monopole antenna was inserted into the transformer tank via the drain valve. UHF signals were acquired and analyzed both in the spectrum analyzer and phase-resolved PD mode. The results of the UHF analysis are shown in Figure 19, which shows the peak and average signal spectra acquired on the UHF antenna.

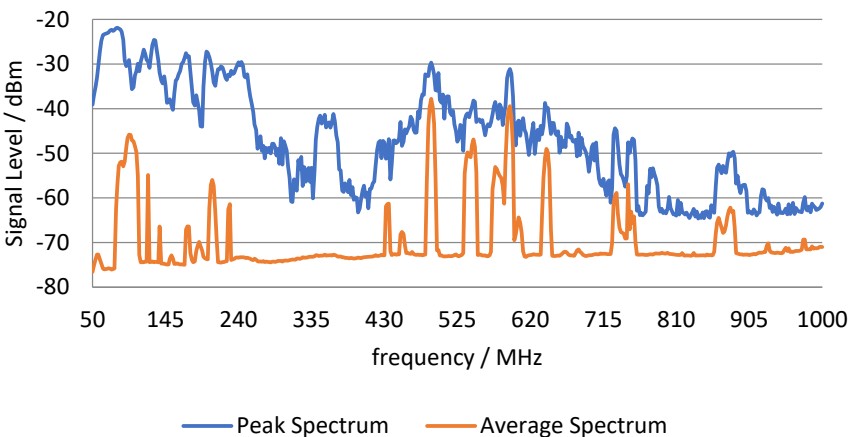

**Figure 19.** Peak and average spectra.

The spectrum analyzer analyzes the signal for peak (blue) and average (orange) signal content across the frequency spectrum, with a dwell time of 40 ms. A signal without spikes or pulses will show very similar peak and average spectra, whereas a signal with fast pulses or spikes will show a significant deviation between peak and average. The spectral areas in which these signals can be observed depend on the type of PD, the signal propagation path, and the measurement circuit used. Spikes in a signal are indicative of PD, and in the present case significant differences between peak and average were observed.

In order to assess whether the pulse-shaped signal characteristics were PD-related, the system was tuned to specific frequencies where peak and average spectra deviate, and narrow-band pulse analysis measurements were performed.

Using the monopole antenna, the frequency at which this specific type of PD was detectable ranged from 50 MHz to about 700 MHz.

### 3.11. On-Site PD Measurement at 1000 MVA, 400 kV Autotransformer

In one of the earliest applications of UHF sensors on a transformer, a 1000 MVA, 400/275 kV autotransformer was fitted with two PTFE windows on the top of the tank during factory repair [31]. On return to service, there was initially no PD. However, after a few weeks in operation, alarms caused by gassing led to the unit being withdrawn from service. Subsequently, the transformer was used as the test object for a major project coordinated by National Grid in the UK, with the aim of comparing various online PD detection techniques. The transformer was energized on-site via its tertiary winding. PD was observed at a modest overpotential by most of the groups applying PD detection to the unit.

UHF sensors S1 and S2 were cabled back to the test cabin using 46 m RG213 cables. A dual preamplifier (26 dB gain) was used to boost signals to a level suitable for capture using a digital sampling oscilloscope (10 GS/s, 3 GHz bandwidth). Based on the time difference of arrival of the UHF signals at S1 and S2, two different PD sources were identified. A pair of UHF signals from the larger of the two PD sources is shown in Figure 20. These showed a consistent time delay of 7.5 ns between arrival at S1 and S2. Taking the first of the two bursts of UHF signal apparent in Figure 20, we have 'views' of the same PD pulse from two different sensor positions on the tank. Considering the signal S1 first, Figure 21a shows the portion of the UHF PD signal selected for FFT processing to the frequency domain. Figure 21b shows a selection of the signal from the same data that represents background noise between the two PD pulses. Figure 22 compares the relative spectral density of the PD and noise sections of signal S1, showing a high signal-to-noise ratio and indicating that most of the signal energy is concentrated in the range of 400–1100 MHz.

amplitude
(0.5 V/div S1; 0.25 V/div S2)

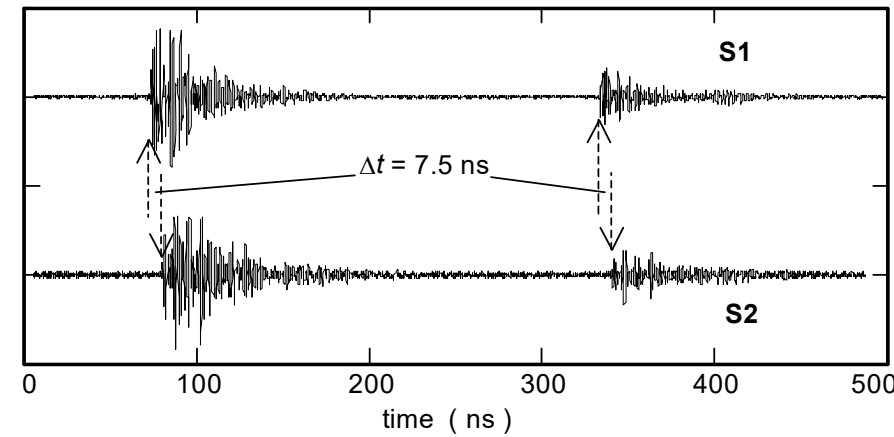

**Figure 20.** Time-domain signals captured at sensors S1 and S2 simultaneously, showing the UHF responses to a PD pulse at around 100 ns followed by a smaller pulse about 250 ns later.

The same procedure followed for sensor S2 (which was several meters away from S1) gives a different perspective on the PD source. Figure 23a shows the portion of the S2 PD signal selected for FFT processing, and Figure 23b shows a section of S2 background noise. Figure 24 compares their relative spectral densities, showing that most S2 signal energy is concentrated in the range of 400–700 MHz.

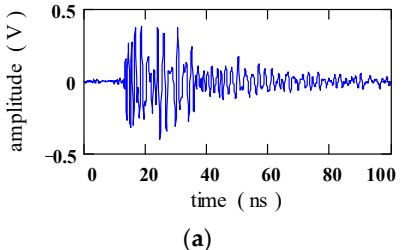

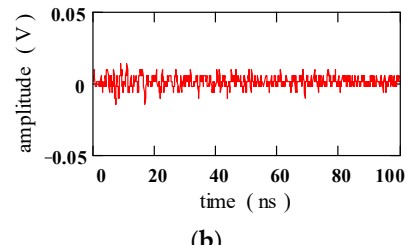

**Figure 21.** (**a**) A 100 ns portion of the UHF PD signal at S1 taken from Figure 20. (**b**) A portion of background signal selected from the same data.

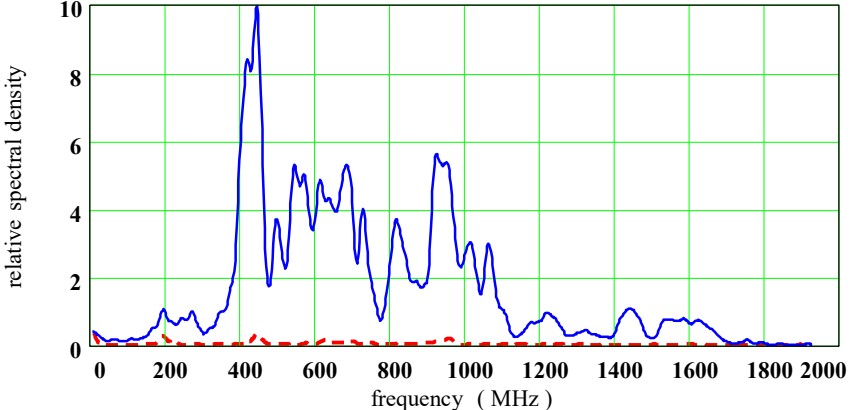

**Figure 22.** Comparison of the computed FFT spectral densities for the two signals in Figure 21.

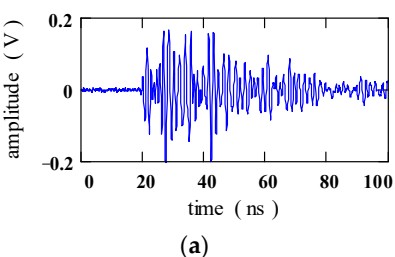

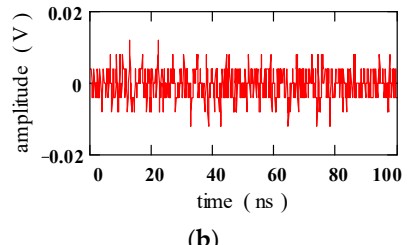

**Figure 23.** (**a**) A 100 ns portion of the S2 UHF signal taken from the signal in Figure 20. (**b**) A portion of S2 background signal selected from the same record.

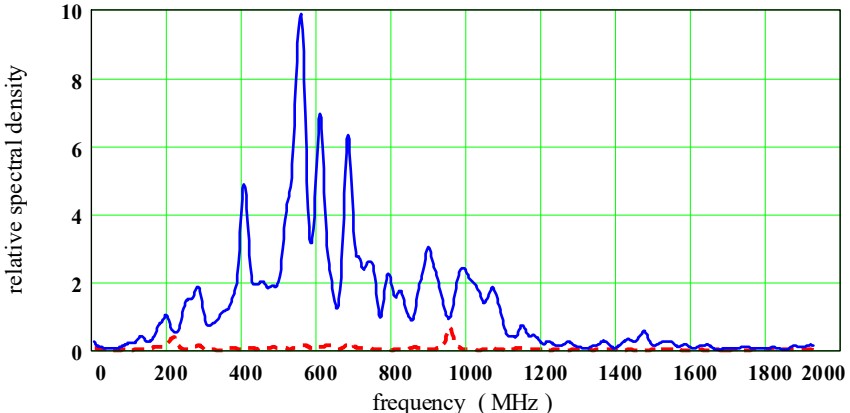

**Figure 24.** Comparison of computed FFT spectral densities for the S2 signals of Figure 23.

## 4. Conclusions

The wide variety of use cases presented in this paper illustrate relevant learnings when UHF measurements are used for PD detection in power transformers.

The literature review of various laboratory measurements revealed a frequency spectrum of UHF PD measurements in the vast majority in the range of 200 MHz to 1 GHz. Newer publications determine the frequency content of the PD pulses in the range of 3 to 6 GHz, which suggests that the emitted frequencies in older publications may be underestimated due to the limited measurement bandwidth of sensors, amplifiers, and measurement systems.

The actual frequency content of UHF PD measurements in power transformers always depends on the specific case and is influenced by the PD source itself, its location inside the transformer, and hence the propagation path to the antenna. Additionally, the frequency of the antenna, damping inside measurement cables, and the bandwidth of the measurement device all determine the resulting PD spectra. However, the entire body of widespread use cases reveals a common frequency range in which all presented measurements provided signal power (see Figure 25) which is approximately in the range of 400 MHz to 900 MHz (minimal common frequency range). The overall frequency range covered is from 50 MHz to 1800 MHz, with peak signal power (meaning highest sensitivity for PD detection) also ranging over a wide frequency range (approximately 100 MHz to 900 MHz). Please note: The frequency ranges provided in Figure 25 are obtained from the individual use cases. Only ranges where the signal power is significantly above the noise level (SNR >> 2) are considered. Peak values are only shown if peak signal power could be derived explicitly from the use cases' spectra.

From these practical observations, a few conclusions can be drawn for UHF PD measurement devices and setups. A UHF PD measurement system should roughly cover the frequency range from approximately 400 MHz to 900 MHz in order to provide basic sensitivity. A wider range is preferable since the actual PD signal peak of an individual case can be outside this range.

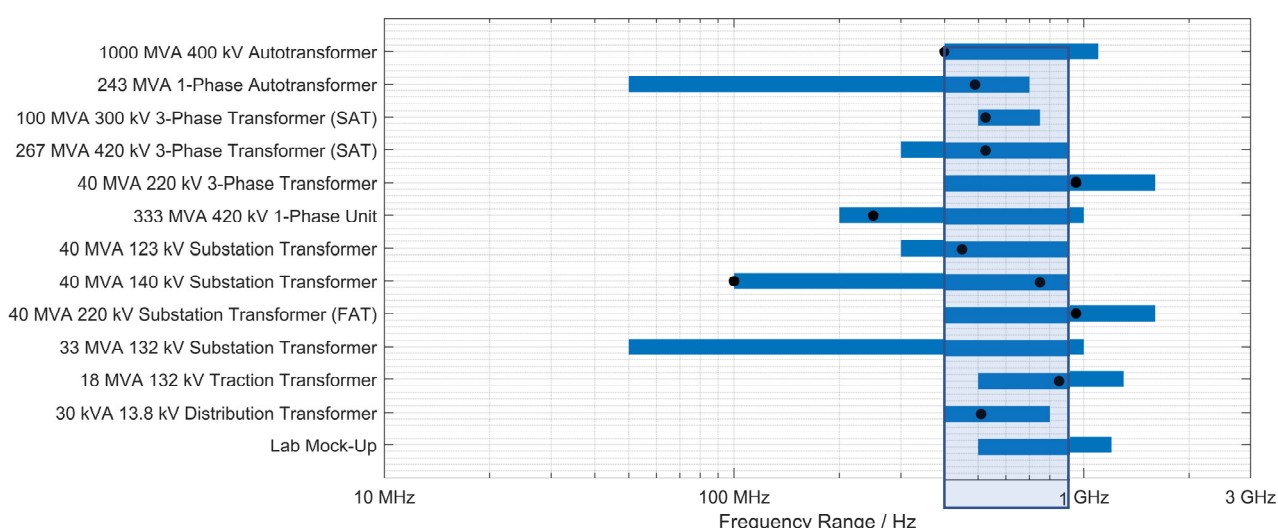

**Figure 25.** Overview of frequency spectra of presented use cases; black dots indicate peak frequency.

**Author Contributions:** Conceptualization, S.T.; methodology, S.T.; investigation S.T., C.P.B., W.S., R.A.S., B.A.d.C., P.F., M.B., M.J., F.W. and M.S.; data curation, S.T., C.P.B., W.S., R.A.S., B.A.d.C., P.F., M.B., M.J., F.W. and M.S.; writing—original draft preparation, Ch. 2 and 3.7: C.P.B., Ch. 2: W.S., Ch. 3.1: R.A.S., Ch. 3.2: B.A.d.C., Ch. 3.10: P.F., Ch. 3.5, 3.8 and 3.9: M.B., Ch. 3.3 and 3.12: M.J., Ch. 3.4 and 3.11: F.W. and Ch. 3.6: M.S.; writing—review and editing, S.T., M.B. and C.P.B.; supervision, S.T. All authors have read and agreed to the published version of the manuscript.

**Funding:** This research received no external funding.

**Data Availability Statement:** Not applicable.

**Conflicts of Interest:** The authors declare no conflict of interest.

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
