# Peer review of "Frequency Range of UHF PD Measurements in Power Transformers"

_energies, doi:10.3390/en16031395_

Round 1

Reviewer 1 Report

The paper presents an analysis of UHF PD measurements on many case studies from laboratory and real transformers. The authors focused their study on the frequency range of interest to maximize the performance of the diagnostics. The paper is well written and the many use cases are very well documented. The only minor comments are editorial: in many places there is a missing space between either a value and a unit, or text and [reference], etc. Also, please make sure to define an acronym only once and at the first appearance in the text (like PD, UHF, etc.).

Author Response

Thank you for your interest in the paper and your valuable comments. We checked the text and improved the space between number and unit and took care regarding the usage of acronyms.

Reviewer 2 Report

The authors have made an interesting paper. Progress in UHF techniques for PD detection in power transformers has been presented. The manuscript also provides valuable updated information on the frequency contents of the PD signals. The important PD parameters of the simulated and actual cases are provided in the manuscript. Also, the references are provided properly. However, I still have some comments and questions as follows.

1)       It seems that the frequency bandwidth of the digital oscilloscope on Page 4 is only 1 GHz. Is it sufficient?

2)       Please provide the reason that the frequency bandwidth of the amplifier on Page 7 of not less than 1 GHz is sufficient in the experiment.

3)       From the questions #2 and 3, the signal with the frequency of over 1 GHz is attenuated. That disagrees with the conclusion which states that the PD signals measured by appropriate and wide frequency bandwidth sensors and measuring instruments have the maximum frequency from 3 GHz to 6 GHz. Please clarify it.

4)       In Fig. 25, please provide the criteria for the determination of the maximum and minimum frequencies. Also, the frequencies of the peak signal powers in some cases are missing. Please clarify it.

5) The abbreviation word (approx.) should be replaced by the full word (approximated).

Author Response

Thank you for your interest in the paper and your valuable questions which allow us to improve the understanding and message of the paper.

1)       It seems that the frequency bandwidth of the digital oscilloscope on Page 4 is only 1 GHz. Is it sufficient?

Both, the lab experiments and the use cases presented in the paper were acquired in a large time frame. Hence, the equipment used for some measurements in the past are outdated by temporary standards. The authors agree that a higher bandwidth (as provided in other uses cases) would be preferable. Yet, the authors decided to include older measurements anyway, if they provided a sound PD analysis. On Page 4 an additional sidenote on the oscilloscope’s bandwidth has been added.

For the MDO 3104 oscilloscope used in the experiment described in Section 2.2, the -3 dB bandwidth for the analog input is ≥1000 MHz and the -6 dB bandwidth is ≥1250 MHz. On the other hand, the main operating frequency band of the Schwarzbeck VULP9118A log-periodic antenna is 200-1200 MHz - throughout this range, VSWR is less than 2. However, if we assume that the acceptable VSWR value is ≤3, then the antenna frequency band extends to the range of ~150-1500 MHz.  

Considering the parameters of the antenna and the fact that among the tested types of PD, only creeping discharges generated signals with a frequency slightly higher than 1 GHz, it can be assumed that the oscilloscope used in the experiment met the minimum requirements for the bandwidth. However, the authors are aware that frequency components above 1000 MHz may have been attenuated. Therefore, in the near future, it is planned to perform more comprehensive laboratory tests of partial discharges in oil-paper insulation using an oscilloscope and antennas covering the entire range of VHF (30-300 MHz) and UHF (300-3000 MHz) bands.

2)       Please provide the reason that the frequency bandwidth of the amplifier on Page 7 of not less than 1 GHz is sufficient in the experiment.

The bandwidth of min. 1 GHz is proposed by Cigre TB 861. The suggestion is driven by the majority of measurements which provide (some) frequency content of PD UHF Signals in the range from a few 100 MHz to approximated 1 GHz (empirical finding). However, a higher bandwidth can be beneficial, e.g. compare to use case 3.2 or 3.4. Additional explanation has been added on page 7.

3)       From the questions #2 and 3, the signal with the frequency of over 1 GHz is attenuated. That disagrees with the conclusion which states that the PD signals measured by appropriate and wide frequency bandwidth sensors and measuring instruments have the maximum frequency from 3 GHz to 6 GHz. Please clarify it.

Hopefully, the additional explanations on pages 4 and 7 clarify it for the reader: a higher bandwidth is beneficial. For practical (and cost) considerations in field use, the bandwidth should not be below 1 GHz, since in there is a high probability, that PD will provide UHF signal power in this frequency range. However, if a larger bandwidth system is available (and within budget), its use is appreciated. 

4)       In Fig. 25, please provide the criteria for the determination of the maximum and minimum frequencies. Also, the frequencies of the peak signal powers in some cases are missing. Please clarify it.

Explanation has been added in the text. Basically, Frequency range is selected if SNR is significantly larger than 2. Yet, for some use cases raw data was not available, hence, the frequency selection had to be obtained optically from the power density spectra shown in the paper. Same applies for the peak values. Peak values only included if they could be explicitly derived from the spectra. 

5) The abbreviation word (approx.) should be replaced by the full word (approximated).

all abbreviations have been changed

Reviewer 3 Report

None.

Author Response

There were no comments and suggestions to deal with.